# Preliminary Results in Testing of a Novel Asymmetric Underactuated Robotic Hand Exoskeleton for Motor Impairment Rehabilitation

**Flaviu Ionuț Birouaș** [1,†] , **Radu Cătălin Țarcă** [1,*,†] , **Simona Dzitac** [2,†] **and Ioan Dzitac** [3,4,*,†]

1   Mechatronics Department, University of Oradea, 410086 Oradea, Romania; fbirouas@uoradea.ro
2   Energetics Department, University of Oradea, 410086 Oradea, Romania; simona@dzitac.ro
3   Mathematics and Computer Science Department, Aurel Vlaicu University of Arad, 310130 Arad, Romania
4   Economic Sciences Department, Agora University of Oradea, 410526 Oradea, Romania
*   Correspondence: rtarca@uoradea.ro (R.C.Ț.); idzitac@univagora.ro (I.D.)
†   All the authors contributed equally to this work.

**Abstract:** Robotic exoskeletons are a trending topic in both robotics and rehabilitation therapy. The research presented in this paper is a summary of robotic exoskeleton development and testing for a human hand, having application in motor rehabilitation treatment. The mechanical design of the robotic hand exoskeleton implements a novel asymmetric underactuated system and takes into consideration a number of advantages and disadvantages that arose in the literature in previous mechanical design, regarding hand exoskeleton design and also aspects related to the symmetric and asymmetric geometry and behavior of the biological hand. The technology used for the manufacturing and prototyping of the mechanical design is 3D printing. A comprehensive study of the exoskeleton has been done with and without the wearer's hand in the exoskeleton, where multiple feedback sources are used to determine symmetric and asymmetric behaviors related to torque, position, trajectory, and laws of motion. Observations collected during the experimental testing proved to be valuable information in the field of augmenting the human body with robotic devices.

**Keywords:** asymmetric underactuated; rehabilitation; robotic exoskeleton; symmetric and asymmetric trajectory; Bowden cable; video processing data

## 1. Introduction

This paper presents research and advances in the field of medical robotics, with a focus on data analysis of the symmetrical and asymmetrical mechanical behavior of the human hand during motor therapy rehabilitation using a novel robotic exoskeleton. As seen in numerous works regarding rigid exoskeleton applications for the human hand [1,2], there are a wide range of applications in this field that include medical [3–5], military, aerospace [6], and industrial uses. The need to augment the human body is driven by recent advances in technology [7,8] and the increasing automation of daily living [9]. Robotic exoskeletons as sometimes seen in movies have transitioned from the science fiction realm to real world applications due to recent advances [10] in a number of multidisciplinary fields such as mechatronics, artificial intelligence, bioengineering, medical robotics, and many more. Robotic exoskeletons continue to advance [11] and soon will become a big part in our daily lives, be it for medical rehabilitation, motor assistance for elder citizens, enhanced strength for military operations, or safer and easier work conditions in the modern factory environment. These types of robotic systems are slowly becoming an integral part of society [12], as a result, human–robot interfacing needs to be researched and understood in detail in order to improve the user experience, efficiency, and design of these devices. The human body is one of the most complex systems to attach to and augment with

robotic devices [13–15]. The biomechanical nature of the human body contains both symmetric as well as asymmetric elements from a geometric point of view [16,17] and also generates symmetric and asymmetric trajectories and behaviors [18,19].

This paper aims to develop a new concept of a rigid robotic exoskeleton that adapts to the symmetric and asymmetric geometry and behavior of the human hand for research purposes in the field of medical robotics. The developed exoskeleton can also potentially be derived and optimized for a series of applications in areas other than medical rehabilitation such as the areas mentioned above.

The goal of the developed device is to provide an improved quality of life for people who suffer from motor impairment disability. The presented work discusses the development and research of a rigid robotic exoskeleton for rehabilitation therapy, mainly for people who had suffered a cerebrovascular accident, also known as stroke. The research data presented covers the development of a new concept of a rigid robotic hand exoskeleton and the symmetrical and asymmetrical behavior of the robot–human interaction during functional testing. The system integration and testing carried out are presented in a comprehensive study based on data generated from video processing software and sensors.

## 2. Materials and Methods

### 2.1. General Considerations

Taking into consideration a multitude of existing developed robotic hand exoskeletons, there are a few design factors that need to be taken into account. There are a number of structural designs that can be implemented in order to transmit motion to human fingers. These designs can be divided into three types, one of which is based on rigid structures that implement classical mechanical actuation. The second one is based on soft robotics [20,21], a new field of robotics that uses various types of soft actuators [22–25], and the third type, is based on hybrid actuation, which implements a combination of rigid, soft, and compliant [26] actuation systems [24,27–29]. In this paper, the structural type used belongs in the rigid structure category. This type of construction using a rigid exoskeleton implies that some parts must be customized for each wearer [30], in other words, a good design would permit interchangeable components to be easily swapped out and reconfigured. One alternative for customizing for each wearer is to use standard sizes similar to clothing and footwear. Having a kit of standardized sizes of elements for the fingers and palm region can speed up the process of configuring the exoskeleton for the wearer.

The device's level of complexity is increased due to the asymmetric distribution of the fingers since each finger has a unique set of anthropometric dimensions. Not only does each finger have unique anthropometric dimensions, but also the phalanges have unique anthropometric dimensions for each of the subjects, following a Fibonacci dimensional ratio [31–33]. As seen in Figure 1, a visual rendering of the human hand mechanical model is presented, where the asymmetric nature of the human hand can be observed. The phalange area represents the finger segments that comprise the thumb, index, middle, ring, and little finger. The blue segments represent the phalanges of the fingers, while the red segments represent the joints of the fingers, where the cylindrical joints have one degree of mobility (DOM) (comprised of 1 rotation) and the universal joint has two DOM (comprised of two rotations).

Considering a variety of mechanical designs present in this field of research [34], there are a few distinct constructive types that stand out. A solution with direct matching of the finger joint centers (DMFJC) was developed by Chiri et al. [6] which can be observed in Figure 2a. This type of design is a good type of construction due to its behavioral similarities to the biomechanics of the human hand. A limitation for this design is that the direct matching of the joints is possible on the Distal Interphalangeal (DIP) and Proximal Interphalangeal (PIP) joints, but mechanically it cannot be implemented on the Metacarpophalangeal (MCP) joint due to the hand anatomy. For the MCP joint, there are a number of other solutions that rely on more complex mechanisms to obtain an actuated or

underactuated movement. The example shown in Figure 2a is an implementation of an underactuated MCP joint using a rotation translation mechanism controlled by a single actuator.

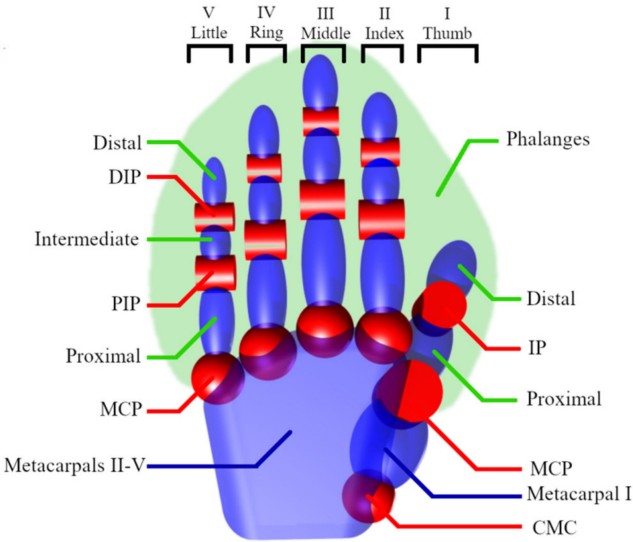

**Figure 1.** Proximal Interphalangeal (PIP), Distal Interphalangeal (DIP) and Metacarpophalangeal (MCP) connected to Metacarpals I joint have one axis of rotation, while the MCPs connected to Metacarpals II-V and Carpo-Metacarpal (CMC) joints have two axes of rotation.

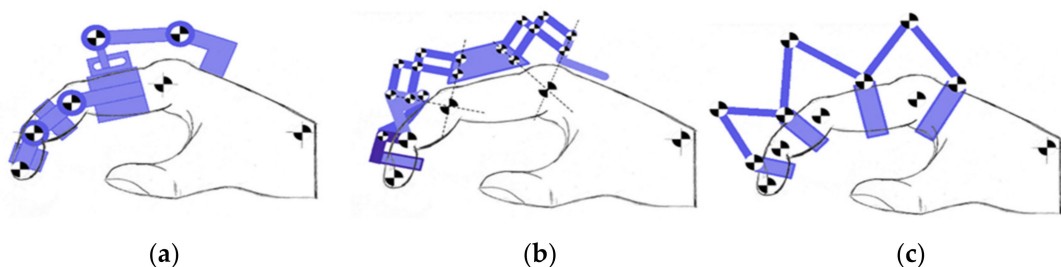

(**a**)        (**b**)        (**c**)

**Figure 2.** (**a**) Direct matching of the finger joint centers by Chiri et al. (**b**) Linkages for remote center of rotation (Shields et al.). (**c**) Underactuated redundant linkage (Wege et al.).

A significant issue that is critical to an exoskeleton mechanical structure is the orthotic shell. One of the most crucial dimensional characteristics is the thickness in the lateral areas of the fingers that is coaxial to the finger joints. This dimension is highly dependent on the wearer and the hand anthropometrics. In other words, a wearer that has thinner and longer fingers will have more space between the fingers, which will permit the mounting of an exoskeleton. In comparison, a wearer that has thicker and shorter fingers will have less space between the fingers, thus resulting in the need for a thinner orthotic shell. The finger segments of the exoskeleton must be designed in such a way as to not hinder the natural motion of the finger or even produce discomfort.

Another type of construction, developed by Shields et al. [35], implements a more complex structure based on a mechanism with linkages for remote center of rotation (LRCR), as seen in Figure 2b. Although it seems like a bulky design, considering the large and complex mechanism, it has the considerable advantage of saving a lot of space between the fingers, an essential factor to take into account when designing the orthotic shell of the exoskeleton fingers.

The hand compliance is also an important factor, so to produce a flexion and extension motion of the biological fingers some designer such as Wege et al. [36] utilized an underactuated mechanism for all joints by implementing an underactuated redundant linkage (URL) structure as seen in Figure 2c. The size of the mechanism is considerably larger than the one implementing a mechanism with the

direct matching of the joints. While this design does not have the precise control of each phalange individually as the mechanism with LRCR structure [35], it has the advantages that it can be operated using fewer actuators and can provide a more natural movement of the wearer's hand due to its underactuated mechanism and the compliance of the biological hand.

*2.2. Cable Drive Transmission*

Cable-driven transmission in the literature of mechanics can refer to more than one type of mechanical transmission [37,38]. In this paper, two types of cable transmission are used, namely pulley-cable transmission [39] and Bowden cable transmission [40]. A Bowden cable is a type of flexible cable used in applications where there is a need to transmit mechanical force or energy, implementing the movement of an inner cable relative to a hollow outer cable housing known as a sheath. In the area of robotics, this form of actuation is usually applied for remote actuation of a robotic joint; force is delivered to the remote joint by means of mechanical displacement between the cable and the outer sheath.

The main factors influencing the cable efficiency are the normal forces on the cable, which are determined by cable tension or preload, the friction coefficients resulting between material combinations, and velocity of the inner wire. Friction between the internal cable and the external sheath usually has an impact on the entire assembly efficiency. Losses and inefficiencies of the Bowden transmission are mainly a result of the complex and non-linear friction phenomena. As described by Kaneko [41], Coulomb friction, viscous friction, stiction, and stick-slip may occur in Bowden cable transmission systems.

The main geometric parameter influencing friction between the sheath and the cable is the total wrap angle of the cable system, illustrated in Figure 3c. A simplified representation of the friction losses of a Bowden cables system can be represented by analogy to sliding a cable over a fixed cylinder at a constant velocity, as indicated in Figure 3a. For this simplified representation, the friction can be expressed by using the expression [42]:

$$\frac{F_{in}}{F_{out}} = e^{-\mu\theta} \tag{1}$$

where:

1.  $F_{in}/F_{out}$ is the ratio of input to output forces,
2.  $\mu$ is the kinetic coefficient of friction between sheath and cable,
3.  $\theta$ is the total wrap angle.

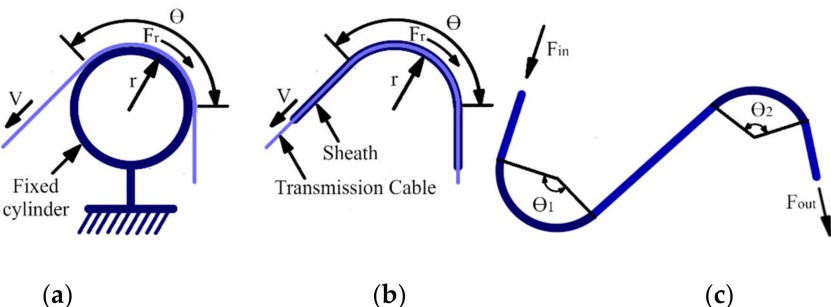

|      (a)      |      (b)      |      (c)      |

**Figure 3.**    (**a**) Simplified equivalent representation of friction in a bowden transmission. (**b**) Representation of friction in a Bowden transmission. (**c**) Total wrap angle $\theta$ of the cable system.

The total wrap angle $\theta$ of the cable system represented in Figure 3c is defined by the equation [43]:

$$\theta = \theta_1 + \theta_2 = \sum_{i=1}^{n} \theta_i \tag{2}$$

In Figure 4, a model for determining the cable tensions at any point along the mechanism of the Bowden cable system is presented. A model based on Coulomb friction is considered since the system does not use lubricants, and the device can be generally considered on a macro scale. The current version of the prototype is designed to develop speeds and torques higher than needed for rehabilitation purposes. As a result, the frictions generated in the system are not a major challenge for the closed-loop control system [44]. However, at some later point in the project's development and optimization, other friction models may be required. Later optimizations such as reducing the scale of the actuation system may require a more thorough study regarding multiple or even more precise friction models [45,46]. For the current research, the work of Kaneko [41] is considered as the starting point for the Bowden cable transmission mechanism model. The normal force originating from the curved sheath creates friction force between the sheath and the cable. The friction force that appears in the mechanism has a nonlinear tension distribution along the wire. The equations that describe this phenomenon can be expressed using the following equations [41]:

$$T(p) = \begin{cases} T_{in}\exp\left(-\frac{<\mu>}{R}p\cdot sign(v)\right) & (p < L_1) \\ T_0 & (L_1 \le p) \end{cases} \tag{3}$$

$$sign(v) = \begin{cases} 1 & (v \ge 0) \\ -1 & (v < 0) \end{cases} \tag{4}$$

$$L_1 = \min\{p \in T(p) = T_0\} \tag{5}$$

$$\begin{aligned} T_{in} &= T(p = 0) \\ T_{out} &= T(p = L) \end{aligned} \tag{6}$$

where:

1. $T(p)$ represents the tension of the cable at position $p$,
2. $\mu$ is the kinetic friction coefficient between sheath and cable,
3. $\theta$ is the summation of the bent angle of each segment,
4. $v$ (noted as $\xi$ in some works) is the velocity of the cable relative to the sheath,
5. $d\gamma$ is the angle subtended by the arc of length $dx$,
6. $R$ is the radius of the sheath curvature,
7. $T_0$ tendon preload,
8. $L$ is the total length of the sheath,
9. $T_{out}$ output tension,
10. $T_{in}$ input tension.

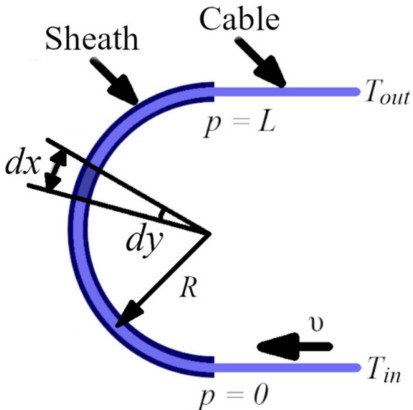

**Figure 4.** Cable tension model parameters of the Bowden cable system.

Numerous studies focus on determining and compensating the friction in a Bowden system. One method utilizes closed-loop feedback of the output tension [47–49]. The compensation is done by monitoring the stress of the cable on the output side, controlling the actuator in such a way that the output tension follows the reference set point. Other studies utilize position-based impedance feedback [41,50,51] combined with a decrease of pre-tension with a slack prevention actuation mechanism to reduce the effect of the friction [52]. For this paper, the friction compensation method relies on the position and current feedback of the system as described in a previous paper [44].

## 3. Results

### 3.1. Mechanical Concept

After observing several mechanical designs, it was determined that the links that comprise the exoskeleton segments corresponding to the finger phalanges do not need to be controlled individually [36]. A better solution is to rely on the body's natural compliance while actuating the exoskeleton. This structure, in turn, generates a natural asymmetric law of motion of the fingers that otherwise would be more difficult to recreate by directly controlling the individual links. As seen in anthropometric studies [53], the human hand's law of motion can have drastic differences from one person to another. According to the asymmetric law of motion, the flexion trajectory and extension trajectory are not symmetrical. As a result, an underactuated [54] mechanical structure is considered in developing the robotic exoskeleton presented and studied in this paper. One actuator is considered for each finger. In Figure 5a representation of the mechanism is given for one finger.

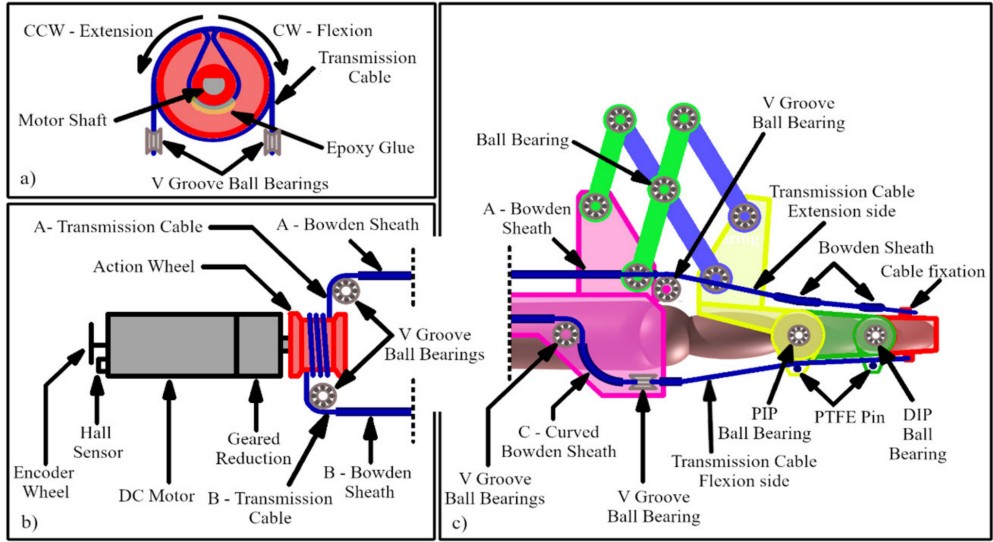

**Figure 5.** Finger actuation mechanism concept. (**a**) Action wheel wire mounting, rotation to translation conversion mechanism, front view. (**b**) Rotation to translation conversion mechanism, top view. (**c**) Cable transmission in the orthotic shell, lateral view.

The actuation is done using a DC motor mounted with a 31:1 ratio gearbox reduction. Position and displacement feedback are achieved using a Hall effect-based quadrature encoder [44]. The motion is transmitted from the motor output shaft (after the gearbox) to the finger, using a mixed transmission mechanism that contains cable-pulley transmission segments and Bowden cable transmission segments, also referred to as tendon-sheath transmission. The angle of the sheath curvature is fixed. As seen in Figure 5a,b, the motor actuates the action wheel, which in turn produces a symmetrical push/pull movement on the transmission cable that is guided further via ball bearing pulleys.

As represented in Figure 5a, the direction of the action wheel rotation produces a pulling motion on one end of the cable, while on the other, it creates a symmetrically pushing motion. The two ends of

the wire are connected further in the mechanism; one end noted as A—Transmission Cable in Figure 5, is guided over the finger to produce the extension movement when tensioned. At the other end, the cable noted as B—Transmission Cable is guided under the finger to provide the flexion movement when tensioned. Most of the direction and angle changes of the translation of the cable is done by implementing ball bearing pulleys. The exception is the region between the two bearings from the flexion side of the cable, noted as C—Curved Bowden Sheath in Figure 5c; here a fixed Bowden sheath is used to guide the cable.

### 3.2. Mechanical Design

The mechanical design of the system is done using CATIA V5 R18. The assembly follows the concept described earlier in the paper. The mechanical concept is applied on all four fingers and taking into consideration a series of anthropometric measurements gathered in previous work [53]. The assembly of the system can be seen in Figure 6 where the distribution of the Bowden cable system can be observed.

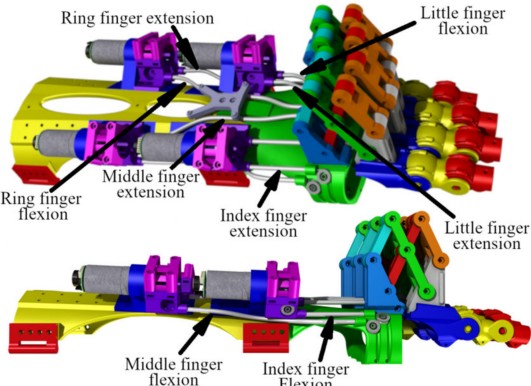

**Figure 6.** Mechanical design of the exoskeleton and the distribution of the Bowden cable transmission.

### 3.3. Manufacturing

The prototype is manufactured using 3D printing technology. In this case, the FDM (fused filament manufacturing) is the preferred 3D printing method. The material used for all parts is standard PLA (polylactic acid). The reason for choosing this material is the fact that it is biodegradable, eco-friendly, and it is easy to use in almost all 3D printers.

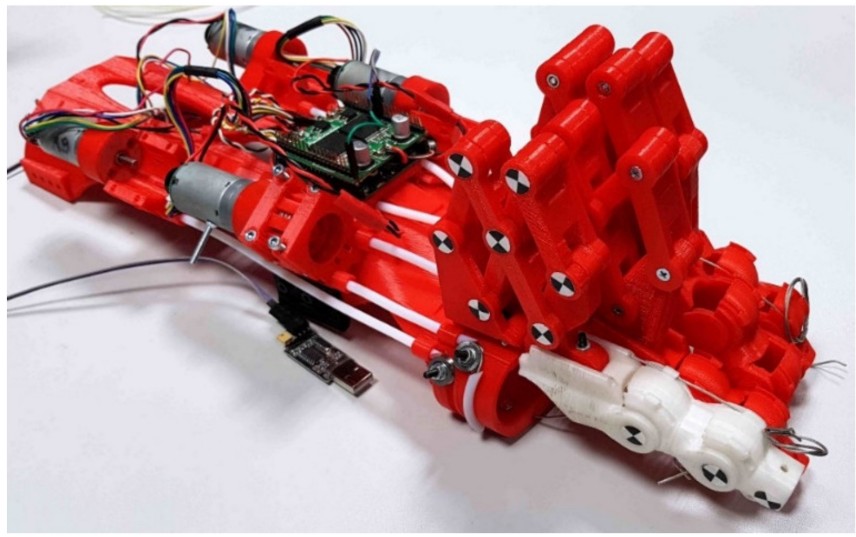

**Figure 7.** 3D-manufactured parts of the exoskeleton assembly.

The manufactured and assembled exoskeleton parts, as seen in Figure 7, were made on three separate 3D printers, two of which were Makerbot clones, while the third was a Whanhao i3 Plus. As with any prototyping process, the design had several iterations due to the optimization of the mechanical design.

### 3.4. Experimental Testing

Testing was carried out using measurements provided by the exoskeleton electronics and by video processing and recognition software. Electronics used for the exoskeleton were specifically designed and manufactured for this application. The data provided by the on-board electronics (as described in previous work) [44] were transferred via RS232 communication protocol onto a PC. The video recognition software was applied to determine trajectories and laws of motion of key points on the exoskeleton. A high-speed camera is required to capture the fast movement of the mechanical components. In the experimental section of this paper, a camera with 240 frames per second was used to capture the video data. The motion of one exoskeleton finger was studied with and without the operator's hand in the device.

### 3.4.1. Exoskeleton Testing without Wearer's Hand

The mechanical behavior of the exoskeleton was analyzed using Kinovea video processing software. The trajectories of key points were tracked via a high frame-rate camera for determining the workspace of the exoskeleton finger. The procedure used to capture and process the video via Kinovea software was described in a previous paper where the software was used to study and determine the anthropometric parameters of the human hand [53]. In Figure 8a the key points and their notations are illustrated. After studying the mechanical behavior of the system, it was observed that the joints rarely moved simultaneously in relation to one another; this phenomenon is simply explained by the friction differences from one joint to another and the friction variable of the cable on the contact guiding surface. As a result of this phenomenon, the joints of the exoskeleton will move sequentially one joint at a time. This phenomenon does not constitute a disadvantage since it offers a good indication for the positions where the system encounters greater frictions and it can be traced back to optimize the mechanism.

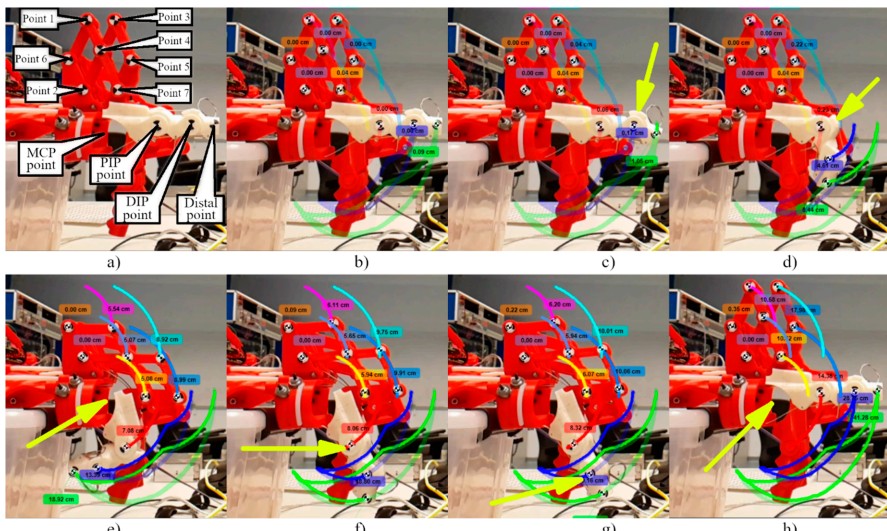

**Figure 8.** Joint phase steps of one flexion/extension cycle. (**a**) Key points and their notations. (**b**) Initial state of the exoskeleton. (**c**) The first phase of actuation—DIP joint flexion movement (**d**) The second phase of actuation—PIP joint flexion movement. (**e**) The third phase of actuation—MCP joint flexion movement. (**f**) The first phase of actuation—PIP joint extension movement. (**g**) The second phase of actuation—PIP joint extension movement. (**h**) The third phase of actuation—MCP joint extension movement.

In Figure 8 the phases of a flexion/extension cycle are detailed. In Figure 8c, it is observed that in the first phase of the finger actuation, the DIP joint is the first to move. The movement of the PIP joint, as seen in Figure 8d, characterizes the second phase. The third phase is the rotation of the MCP remote point, as seen in Figure 8e. The behavior observed is as expected, since the DIP and PIP joints each have two ball bearings, while the MCP remote center of rotation utilizes a more complex mechanism with multiple joints that will inherently encounter more significant friction forces. Based on the same logic, the extension's phase steps will be dependent on each joint's friction. As seen in Figure 8f, the first joint in the sequence to move is the PIP joint, followed by the DIP, as seen in Figure 8g. The final movement is again the MCP remote center of rotation, as seen in Figure 8h. An essential aspect to point out is that although the order of the joint movements for this cycle of flexion/extension is as expected, the system is still an underactuated mechanism. Given the right conditions, the order of the joint movements will not always be the same. Small variations of friction from the joints or cable can result in a different order in the joint movements, producing an asymmetric trajectory cycle during flexion/extension exercises. The next step in the analysis is generating the trajectory of the key points and the exoskeleton's workspace. Based on the captured motion of the key points over several flexion/extension exercises, the workspace seen in Figure 9 is generated.

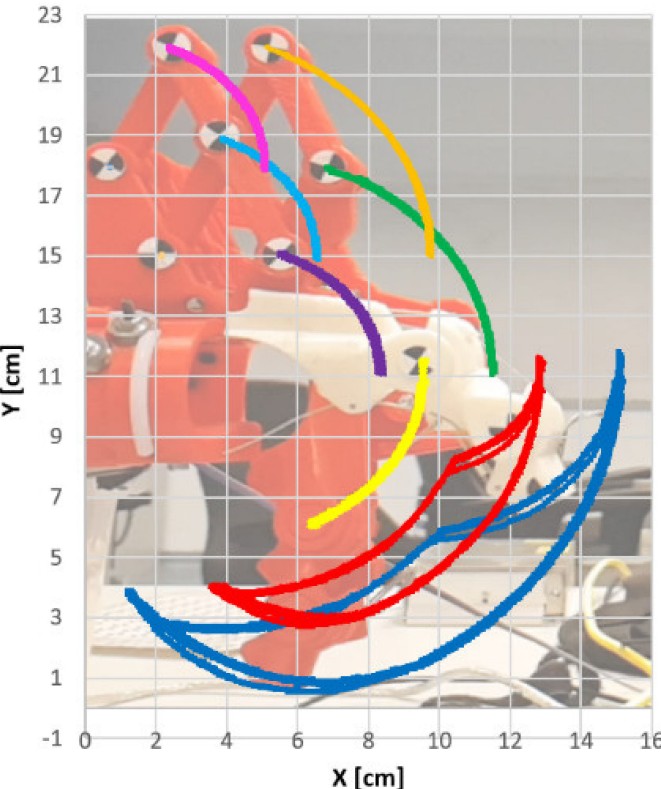

**Figure 9.** Exoskeleton key point asymmetric trajectory cycle and workspace analysis without operator's hand.

A more detailed analysis of the mechanical behavior is illustrated in Figure 10a, where the processed video data is used to generate a graphical representation of the laws of motion for each key point on the X and *Y*-axis.

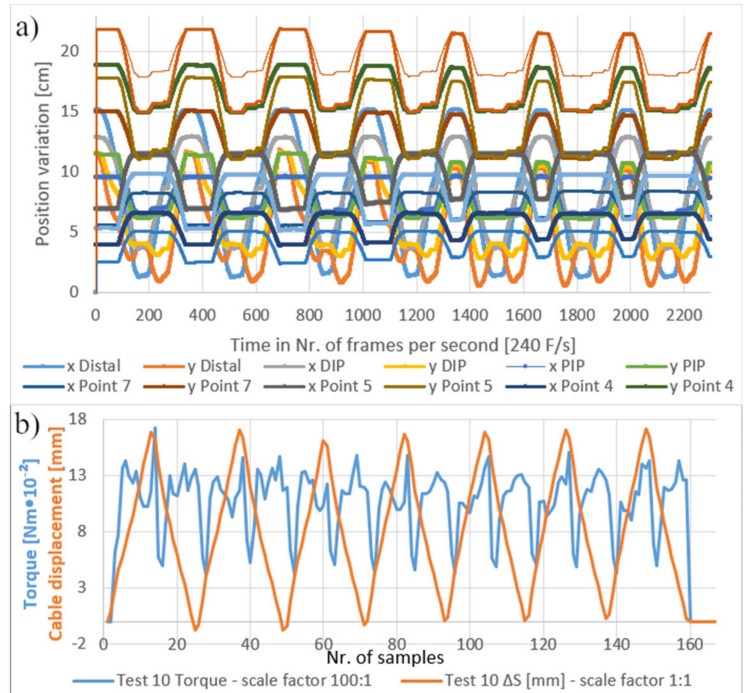

**Figure 10.** (**a**) Graphical representation of the law of motion of each key point when the wearer's hand is not mounted on the exoskeleton. (**b**) Influence of the joints' actuation order on the law of motion and torque.

The chosen example for generating the graph in Figure 10a, confirms that not all flexion/extension cycles have the same order of actuation of the joints when the human hand is not interfaced in the exoskeleton. It is observed that the first four cycles follow the sequence of steps described in Figure 8, while the next three cycles show a noticeable difference, which is the result of a different order of the joints' actuation. The difference is at the flexion part of the cycle, where the first four cycles follow the order DIP, PIP for flexion, and then MCP, while the remaining cycles follow the order DIP, MPC, and then PIP. This change in the joints' order of actuation is also observed in torque, as seen in Figure 10b, where the torque is significantly smaller after the change in the joints' actuation order.

### 3.4.2. Exoskeleton Testing with Wearer's Hand

Similar to previous tests, a trajectory was determined for the key measurement points. In this test, the human hand is interfaced with the exoskeleton. The generated path can be observed in Figure 11, where seven flexion/extension cycles were used to generate the data. An important observation is that, compared with the previous experiments, the mechanical behavior, such as the order of joint actuation, has changed. The first thing that was noticed is that the trajectory of the key points has changed in a way that the flexion/extension exercises tend to produce a more symmetrical pattern with the wearer's hand in the exoskeleton. While experimenting without the wearer's hand in the exoskeleton, the joint actuation was generally done one joint at a time. With the wearer's hand in the exoskeleton, the joints are actuated simultaneously, most of the time. This behavior is due to the reaction forces and friction forces introduced in the system by the biological finger, and its interactions with the exoskeleton's orthotic shell. In Figure 12a detailed graphical representation of the law of motion of the key points with the operator's hand in the exoskeleton is presented.

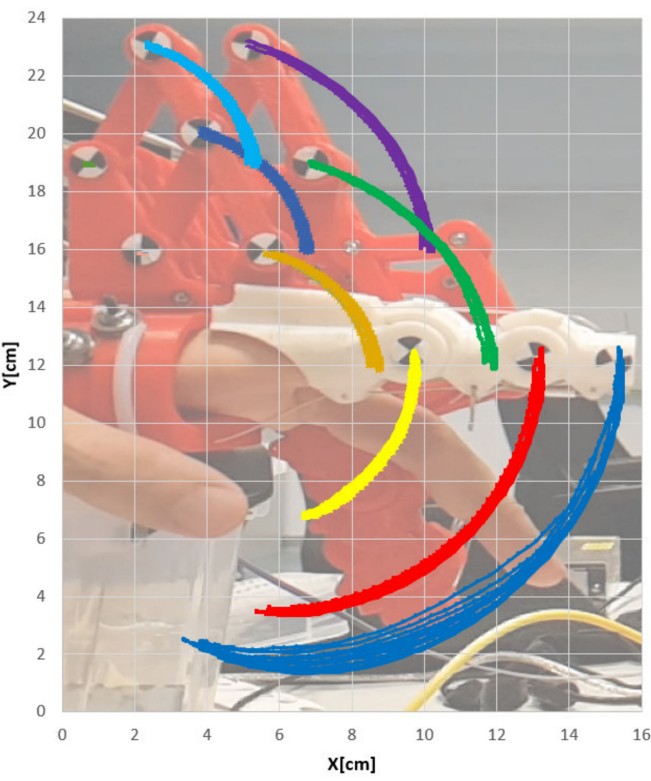

**Figure 11.** Exoskeleton key point trajectory analysis with operator's hand.

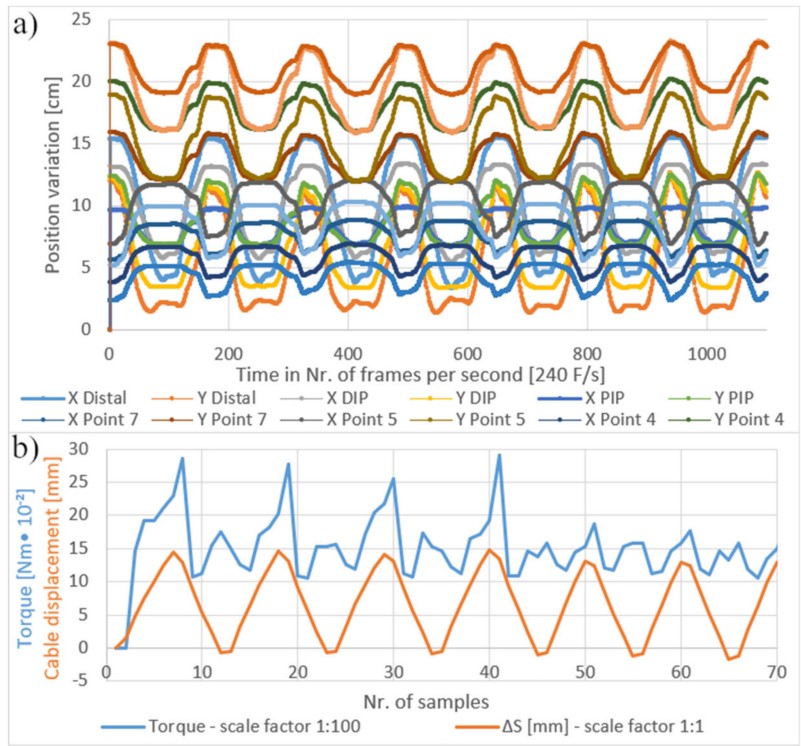

**Figure 12.** (**a**) Graphical representation of each key point's law of motion, while the operator's hand is mounted. (**b**) Torque variations as resulted from operator finger alignment in the exoskeleton.

A comparison of the cable displacement and torque is made for the seven cycles of flexion/extension, as seen in Figure 12b. Another interesting phenomenon that appeared in this experiment is that the torque values measured decreased after a random number of cycles, as seen in Figure 12b;

this phenomenon was investigated further to determine the cause. After extensive experimenting and comparing the data, it was observed that this phenomenon happens in almost every experiment iteration with the wearer's hand in the exoskeleton.

As seen in Figure 13, the torque and also cable displacement start to display a more regular pattern after 200 measurement samples. Extensive testing concluded that the decrease in torque occurs due to the finger self-alignment in the orthotic shell after several flexion/extension cycles. Although the mechanism functions, the biological articulation of the finger and the mechanical joints of the exoskeleton do not match perfectly at the start of exercise, due to coaxial offsets. This phenomenon produces an asymmetric behavior at the start of the exercises and after the biological and mechanical axis auto-align, the behavior tends to become symmetrical. This logical explanation corresponds to the irregular behavior of the system at the beginning and during the first flexion/extension cycles.

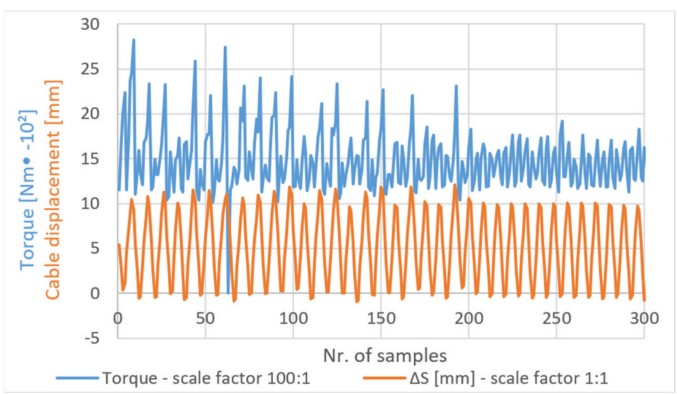

**Figure 13.** Torque and cable displacement stabilization due to the biological finger self-alignment in the orthotic shell of the exoskeleton.

## 4. Discussion

The proposed new exoskeleton design has been developed, starting from a thorough analysis of existing exoskeletons. The device is aimed at solving the mechanical problem in a way that adapts to the geometrical and behavioral parameters of the biological hand. The analysis of a variety of designs based on a rigid mechanical structure present in this field of research [1,4,5,20,27,28,45], showed some remarkably distinct types of construction. The solution developed by Chiri et al. [6], considering the direct matching of the finger joint centers, is a good one, due to its compliance with the asymmetric behavior of the human hand's biomechanics. A limitation for this design appears at the MCP joint, where direct matching of the joint cannot be implemented due to the hand anatomy. In most designs of hand exoskeletons, the DIP and PIP joints are easy to actuate via an exoskeleton mechanism, while the MCP joint presents a bigger challenge to actuate properly. For the MCP joint, there are several alternative solutions, based on more complex mechanisms for generating actuated or underactuated movement. Another type of construction, developed by Shields et al. [35], implements a more complex structure based on a mechanism with linkages for remote centers of rotation. Although it seems like a bulky design, considering the large and complex mechanism, it has a huge advantage considering that it saves a lot of space between the fingers, an important design factor in developing a compliant orthotic shell for the exoskeleton fingers. Comparing the design proposed in this work to the actual state of the art in the field, a series of advantages and disadvantages can be identified. In the first place, it has to be mentioned that the device presented in the paper benefits from the advantages offered by an underactuated mechanism, as a result of a low number of actuators. Another advantage of the proposed device is due to the fact that the actuators are not placed on the fingers' orthotic shell. Therefore, this device has the ability to present easily changeable components based on the person's anthropological measurements. Another advantage resides in a more compact design for DIP and PIP joints, compared to other underactuated models (as well as some fully-actuated models) such as those

of Shields et al. [35], Wege et al. [36], and Zhang et al. [55]. One can mention, among advantages, that the proposed design offers a specified center of rotation for the MPC joint, in contrast to the solutions proposed by Chiri et al. [6]. It can be said that the current design can offer a natural movement of the wearer's hand, compared to other fully-actuated exoskeleton models, such as those presented in Zhang et al. [55], where each joint is independently actuated, thus involving a highly complicated control algorithm, due to the fact that the exoskeleton does not adapt very well to different anthropological dimensions and behaviors. On the other hand, one can mention as disadvantages the lack of certainty for each of the joints' positions, which obviously is a negative characteristic. Even though numerous advantages of the designed device are evident, a notable disadvantage is still present in this solution: the mechanism with the remote center of rotation implemented on the MPC joint is still a relatively large one. Further research has to be undertaken to optimize the device's dimensions. It was noted that the new exoskeleton adapts itself to the wearer's fingers, thus producing a compliant actuation, because of the underactuated mechanism implemented in the exoskeleton design. A notable difference in testing the exoskeleton with and without the wearer's hand is that while the device is being used with the wearer's hand, it produces trajectories that tend to be symmetrical between flexion and extension cycles. In contrast, in testing without the hand, the exoskeleton tends to produce asymmetrical flexion/extension cycles.

## 5. Conclusions

Based on the research results presented in this paper, the functionality of the proposed robotic exoskeleton device is tested and studied. Observations such as mechanical symmetrical and asymmetrical behavior determined in this paper will further benefit research regarding the augmentation of the human body with robotic exoskeletons. The study presented contains valuable data and observations that can be used not only for medical rehabilitation applications, but can potentially extend to other areas of application such as civil use, industrial or home environment, or even military and aerospace applications. Even though the study presented in this paper consists of only initial tests of the exoskeleton prototype, the system proved itself to be a good platform for experimental research. As for the development level of the device, it can be approximated that the device is still a prototype. The final aim of the project is to develop an adequate design for a commercial product. There is plenty of space for future research regarding optimization, such as, for example, size reduction, implementation of soft actuating systems, compliant systems, or hybrid systems, as well as studies related to robustness and device resilience [56]. In future research, we plan to extend the device's functionality, adding upgrades, including more feedback sources and integration of electromyography (EMG) signals with human brain interfaces to monitor rehabilitation progress during automated occupational therapy procedures. Other future updates are oriented towards increasing the device's autonomy in familiar environments, considering integrating elements such as batteries, internal data storage, wireless communication, and easy-to-use user interface software. Although the friction model related to the actuating system presented in the paper is a good starting point in research related to the proposed device, friction compensation control and optimization are broad subjects in themselves [45,46]. They deserve a dedicated and complete study, which may be taken into account in future developments of this project.

**Author Contributions:** Conceptualization, F.I.B. and R.C.Ț.; methodology, R.C.Ț. and I.D.; video processing, I.D. and S.D.; investigation, I.D. and R.C.Ț.; validation, S.D.; design of the CAD model, F.I.B and R.C.Ț.; formal analysis, R.C.Ț.; data curation, S.D.; writing—original draft preparation, F.I.B.; writing—review and editing, S.D. and I.D.; review, proofreading and improvements of the paper, I.D. and S.D.; resources and materials F.I.B.; visualization, S.D. and I.D.; supervision and project administration, R.C.Ț. All authors have read and agreed to the published version of the manuscript.

**Funding:** This research received no external funding. And the article processing charge (APC) is supported by Cercetare Dezvoltare Agora (R&D center of Agora University of Oradea).

**Acknowledgments:** The authors would like to thank the Ph D School of Engineering Sciences—University of Oradea, Romania, for providing technical support and access to online academic databases.

**Conflicts of Interest:** The authors declare no conflict of interest.

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
