# Peer review of "Preliminary Results in Testing of a Novel Asymmetric Underactuated Robotic Hand Exoskeleton for Motor Impairment Rehabilitation"

_symmetry, doi:10.3390/sym12091470_

Round 1

Reviewer 1 Report

A detailed presentation of an exoskeleton prototype design and testing.

Authors better detail how trajectories and motion laws of key points were determined with the video recognition software.

A video of the exoskeleton motion would be extremely useful to better observe the mechanical behaviour analysis with Kinovea-based software. The sequential-based motion of the joints is a disadvantage?

Discuss some of the pros and cons of this exoskeleton, compared with similar prototypes.

Which electronics modules have been used for logging data from the exoskeleton?

How is the torque measured, what kind of sensors do you use.

Check for some typo, see (4) with 'sing' instead of 'sign'.

Conclusions must be point out fewer general aspects and potential future applications and more specific research findings. Some of this design's disadvantages would make outstanding proof of literature awareness.

Author Response

Response to Reviewer 1

[comment 1] Authors better detail how trajectories and motion laws of key points were determined with the video recognition software.

Response: Very good observation, we did not realize that results were somewhat out of context without an explanation related to how the video data acquisition is done, we have published the process in a previous paper regarding anthropometric measurements using video processing. We did not highlight this clearly in the paper and have updated with a clear reference to the paper where the procedure was described. [at line: 252 - 254] In addition, some aspects related to the software are not detailed since it was considered by the authors that the video recognition software in itself is an open source software, and it has extensive video tutorials and community that provide information related to its use.

[comment 2] A video of the exoskeleton motion would be extremely useful to better observe the mechanical behaviour analysis with Kinovea-based software.

Response: Absolutely correct, we forgot that we are able to include videos directly on submission. We are providing two videos of the experiments done with and without the wearer’s hand. We are also including two private links to the videos to be viewed on Youtube:

https://youtu.be/NMdKzdQg3wU

https://youtu.be/a9g4nrTNe0o

[comment 3] The sequential-based motion of the joints is a disadvantage?

Response: Thank you for addressing this question. Not necessarily, but it is an observation related to the mechanism behavior which occurs when the exoskeleton is not mounted on the wearer’s hand. However, it provides an indication to where the system encounters the most friction. We updated and mentioned this at line 259 – 261.

[comment 4] Discuss some of the pros and cons of this exoskeleton, compared with similar prototypes.

Response: Considering the suggestion, we have expanded the Discussion section of the paper with a more in-depth comparison regarding pros and cons. Changes are highlighted using the Track-Changes tool included in Microsoft Word.

[comment 4 and 5] Which electronics modules have been used for logging data from the exoskeleton? How is the torque measured, what kind of sensors do you use.

Response: The electronics modules are custom designed and manufactured for the exoskeleton. These aspects and the data logging related to torque measurements and cable displacement are the subject of a previously published paper that is cited. Reading the review, we realized that the article needs to be more explicit in providing information related to electronics and some data logging, which are presented in a previously published paper. We have made changes to highlight this aspect. [at line: 187, 188, 240 - 243]

[comment 6] Check for some typo, see (4) with 'sing' instead of 'sign'.

Response: The authors have analyzed the whole paper and have updated the typos.

[comment 7] Conclusions must be point out fewer general aspects and potential future applications and more specific research findings. Some of this design's disadvantages would make outstanding proof of literature awareness.

Response: The suggestions are taken into account, and the conclusions have been expanded, taking into account the feedback. Changes are highlighted using the Track-Changes tool included in Microsoft Word.

Reviewer 2 Report

The authors present a very interesting underactuated robot hand exoskeleton.

The paper is well written, technically sounds and to the best of knowledge its content is original and the topic is in the interest of the journal. The result presented are supported by the experimental tests.

The only observation I can make is that authors do not take advantage of some works that I conseder very interesting:

- Giuseppe Carbone1, Cesare Rossi and Sergio Savino, Performance Comparison Between FEDERICA Hand and LARM Hand, International Journal of Advanced Robotic Systems 2015, 12:90 | doi: 10.5772/60523

- Zappatore, G.A., Reina G., Messina, A., Analysis of a Highly Underactuated Robotic Hand, International Journal of Mechanics and Control, Vol. 18, No. 02, pp. 17-23, 2017

Finally I have a suggestion for the authors: if thay are interested in developing novel prothotypes, thay shoud try to reduce the dimension of the whole mechanism.

In conclusion, in my opinion, the paper should be accepted for pubblication.

Author Response

Response to Reviewer 2

[comment 1] The authors present a very interesting underactuated robot hand exoskeleton.

The paper is well written, technically sounds and to the best of knowledge its content is original and the topic is in the interest of the journal. The result presented are supported by the experimental tests.

Response: Thank you very much for the kind words of appreciation for our work.

[comment 2] The only observation I can make is that authors do not take advantage of some works that I conseder very interesting:

- Giuseppe Carbone1, Cesare Rossi and Sergio Savino, Performance Comparison Between FEDERICA Hand and LARM Hand, International Journal of Advanced Robotic Systems 2015, 12:90 | doi: 10.5772/60523

- Zappatore, G.A., Reina G., Messina, A., Analysis of a Highly Underactuated Robotic Hand, International Journal of Mechanics and Control, Vol. 18, No. 02, pp. 17-23, 2017

Response: The suggestions provided are of significant value and have been mentioned in the General considerations section of the paper. [at line: 71]

[comment 3] Finally I have a suggestion for the authors: if thay are interested in developing novel prothotypes, thay shoud try to reduce the dimension of the whole mechanism.

Response: The observation is valuable and will be mentioned in the conclusion for further development.

[comment 4] In conclusion, in my opinion, the paper should be accepted for pubblication.

Response: Thank you very much for your valuable feedback that helped us improve this manuscript.

Reviewer 3 Report

Comment 1:

Analysis of relevant works is not enough. There are several works on using a compliant mechanism concept to develop such a device, e.g., "Topology optimization of a fully compliant prosthetic finger: design and testing", https://ieeexplore.ieee.org/document/7523766. The authors need to compare their work with the literature. They need to first classify the design, e.g., into the compliant mechanism and rigid body mechanism and hybrid. The compliant mechanism design is very useful to designing the so-called soft robotic systems for this application, and it can provide a systematic way to design such a system. I believe it may be a future effort by the authors. It is worth mentioning a methodology for computational design of hybrid mechanisms, see “Towards a Unified Design Approach for both Compliant Mechanisms and Rigid-Body Mechanisms: Module Optimization. https://www.researchgate.net/publication/281664517_Toward_a_Unified_Design_Approach_for_Both_Compliant_Mechanisms_and_Rigid-Body_Mechanisms_Module_Optimization

Comment 2:

The friction is very important, as the authors mentioned. However, the justification of the friction model used needs to be given. In literature, there are comparison of five popular friction models, and that would be a starting point for the authors to give a rational selection of the friction model for their system.

Comment 3:

The title: “Preliminary Testing of a Novel Asymmetric Underactuated Robotic Hand Exoskeleton for Motor Impairment Rehabilitation” is a little bit confusion. It makes no clear about the contribution of the work. Testing can never be a contribution, and testing is employed to test the novel design. A novel design could be contribution.

Comment 4:

How about the reliability, robustness and resilience of the device? The definition of the three concepts can be found from the paper “towards a resilient manufacturing system” (CIRP Annals, 2011). It is interesting to study these three concepts as the device is related to biological system, human hands in this case.

Author Response

Response to Reviewer 3

[comment 1] Analysis of relevant works is not enough. There are several works on using a compliant mechanism concept to develop such a device, e.g., "Topology optimization of a fully compliant prosthetic finger: design and testing", https://ieeexplore.ieee.org/document/7523766.

Response: Thank you for the suggestion. The cited paper is relevant and of significant importance, the work has been referred in the General considerations section of the paper.

The authors need to compare their work with the literature. They need to first classify the design, e.g., into the compliant mechanism and rigid body mechanism and hybrid.

Response: Thank you for the observation. The classification of the design is briefly stated in the introduction; however, we realize that this classification must be highlighted in the introduction more clearly to give a better first idea to the reader. We have made changes to underline this aspect at lines 50, 57, 59 of the manuscript. Classification is also presented in the General considerations section of the paper; however, as pointed out in the review, in the literature classifications of the article the hybrid mechanism classifications type was omitted and has been since updated with the appropriate literature suggested to provide a complete overview as seen highlighted at line 68 - 72 of the manuscript.

The compliant mechanism design is very useful to designing the so-called soft robotic systems for this application, and it can provide a systematic way to design such a system. I believe it may be a future effort by the authors.

Response: The feedback is on point for future optimizations of the exoskeleton and is now added at the conclusion section to be considered for future directions of the project. Also, coincidently it is a good observation for another device the authors are working on based on soft robotics, but as of yet not published due to patenting related constrictions.

It is worth mentioning a methodology for computational design of hybrid mechanisms, see “Towards a Unified Design Approach for both Compliant Mechanisms and Rigid-Body Mechanisms: Module Optimization.

https://www.researchgate.net/publication/281664517_Toward_a_Unified_Design_Approach_for_Both_Compliant_Mechanisms_and_Rigid-Body_Mechanisms_Module_Optimization

Response: The paper mentioned is relevant to the classification of the design, and has been included to broaden the understanding of the classification types.

[Comment 2] The friction is very important, as the authors mentioned. However, the justification of the friction model used needs to be given. In literature, there are comparison of five popular friction models, and that would be a starting point for the authors to give a rational selection of the friction model for their system.

Response: Perfectly pointed observation, we omitted to mention the reasoning behind the decision of using the specific friction model in the manuscript. We have added the relevant information in a short and compressed way, highlighted at line 156 - 165 of the manuscript. The observation is very useful and has also been mention in the conclusion section as well for future objectives due to the fact that the subject in itself has potential to be a vast and comprehensive study that deserves a dedicated research paper.

[Comment 3]: The title: “Preliminary Testing of a Novel Asymmetric Underactuated Robotic Hand Exoskeleton for Motor Impairment Rehabilitation” is a little bit confusion. It makes no clear about the contribution of the work. Testing can never be a contribution, and testing is employed to test the novel design. A novel design could be contribution.

Response: The observation is relevant, and we agree that there can be some confusion taking into consideration the title alone. However, we consider that by mentioning the word “Novel”, we are suggesting that the device is of an original contribution, and by using the combination “preliminary testing,” we are suggesting that the novel device is at its early stages of development and the paper is presenting the findings resulted from preliminary testing.

We consider that the title describes accurate to the content of the paper, and the overall theme is presented. As of its current form, we think the title respects the guidelines of the Manuscript Preparation General Considerations of the journal and that the title is informative and relevant (conforming to review criteria according to Publons Academy).

However, the observation is valid, and we welcome any improvements by suggestions of alternative formulations.

We have considered a small modification of the title to be updated to: “Preliminary Results in Testing of a Novel Asymmetric Underactuated Robotic Hand Exoskeleton for Motor Impairment Rehabilitation.

[Comment 4]: How about the reliability, robustness and resilience of the device?

Response: The device is a prototype, as mentioned in the paper. It is not a product ready for mass manufacturing. At this stage, factors that determine the life expectancy of the device, such as robustness and resilience are not analyzed. Of course, these factors are important at a later stage of the project development where optimizations are considered. For this paper, we are interested in the earlies findings of a new prototype.

The definition of the three concepts can be found from the paper “towards a resilient manufacturing system” (CIRP Annals, 2011). It is interesting to study these three concepts as the device is related to biological system, human hands in this case.

Response: I agree that the concepts stated in the mentioned paper have significant importance in the product life cycle of a device that is meant to go under mass production, however at this early stage of the project and on the content of this particular paper we consider it to be out of scope. But we are considering this suggestion in the conclusion, mentioning that it should be considered at a later stage of development.

Round 2

Reviewer 3 Report

I am happy with the revision.